# Physiological, Performance, and Oxidative Stress Responses to High-Intensity Uphill and Downhill Interval Training

**DOI:** 10.3390/jfmk10040460

**Published:** 2025-11-24

**Authors:** George Theofilidis, Gregory C. Bogdanis, Antonia Kaltsatou, Konstantina P. Poulianiti, Georgia I. Mitrou, Clara Suemi da Costa Rosa, Kalliopi Georgakouli, Antonios Stavropoulos-Kalinoglou, Argyro A. Krase, Fani Chasioti-Fourli, Nikolaos Syrmos, Giorgos K. Sakkas, Yiannis Koutedakis, Christina Karatzaferi

**Affiliations:** 1Experimental Physiology & Myology and Lifestyle Medicine Laboratory, CREHP, DPESS, School of Physical Education Sports Science and Nutrition, University of Thessaly, Karyes, 42100 Trikala, Greece; gtheofilidis@uth.gr (G.T.); akaltsat@gmail.com (A.K.); kpoulianiti@uth.gr (K.P.P.); geomi@hotmail.com (G.I.M.); clarasuemi@hotmail.com (C.S.d.C.R.); kgeorgakouli@uth.gr (K.G.); a.stavropoulos@leedsbeckett.ac.uk (A.S.-K.); arg.krase@gmail.com (A.A.K.); physiologyexperimental@gmail.com (F.C.-F.); gsakkas@pe.uth.gr (G.K.S.); y.koutedakis@pe.uth.gr (Y.K.); 2School of Physical Education and Sports Science, National and Kapodistrian University of Athens, Ethnikis Antistasis 41, 17237 Dafni, Greece; gbogdanis@phed.uoa.gr; 3Bioscience Institute, (UNESP) Sao Paulo State University, Rio Claro 13506-692, SP, Brazil; 4Department of Nutrition and Dietetics, School of Physical Education Sports Science and Nutrition, University of Thessaly, Karyes, 42100 Trikala, Greece; 5Carnegie School of Sport, Leeds Beckett University, Headingley Campus, Leeds LS6 3QT, UK; 6General Hospital of Volos, 38222 Volos, Greece; milanako76@yahoo.gr; 7Faculty of Arts, University of Wolverhampton, Ring Rd, Wolverhampton WV1 1RY, UK

**Keywords:** interval training, endurance performance determinants, REDOX status, downhill running, uphill running

## Abstract

**Objectives**: We examined how opposing running slopes can modulate interval training effects on aerobic performance and reduction–oxidation (REDOX) determinants. **Methods**: Fourteen physically active volunteers, assigned to either the Uphill group (UG) or the Downhill group (DG), completed 16 workouts of ten 30-s runs, at either +10% or −10% grade, with a work-to-rest ratio of 1:2 at 90% of their Maximum Aerobic Speed (MAS) over 8 weeks. Maximal oxygen uptake (VO_2max_), MAS, Running Economy (RE), time to exhaustion at MAS (Tmax), respiratory exchange ratio (RER), and blood lactate at rest, 5th, and 10th runs were evaluated pre-, mid-, and post-training. Also, REDOX markers [Total Antioxidant Capacity (TAC), Protein Carbonyls (PCs) were assessed in blood samples taken at rest and 3 min post-exercise of the first and last workouts. **Results:** VO_2max_ was unchanged in both groups; in the DG, MAS increased (from 14.2 ± 1.7 to 15.0 ± 1.5 km/h, d = 0.43), and post-training RER significantly increased (from 1.06 ± 0.07 to 1.12 ± 0.03). In the last training session, blood lactate levels increased in the UG (from 9.30 ± 2.69 mmοl/L to 13.34 ± 4.64 mmοl/L) but remained low and unchanged in DG (<2 mmοl/L). Post-training, resting TAC decreased in both groups, and the exercise-induced rise in PC levels was attenuated. **Conclusions**: Despite the brief intervention, VO_2_max levels were maintained in both groups, with divergent changes in metabolic, REDOX, and performance indicators; uphill HIIT may serve for enhancing lactate tolerance, while downhill intermittent running may improve running economy.

## 1. Introduction

High Intensity Interval Training (HIIT) is a popular and widespread training modality for a range of exercise participants, whether athletes, recreational, or health-minded individuals, as indicated by global fitness trends [1,2].

Aerobic performance determinants include physiological and functional indicators such as the maximal oxygen uptake (VO_2max_), the Maximum Aerobic Speed (MAS) which is the speed attained at VO_2max_, Running Economy (RE) which reflects oxygen use at a steady-state speed (VO_2_ mL·kg^−1^·min^−1^), and time to exhaustion at MAS (Tmax) which is associated with an overall faster speed in endurance running events and an improved lactate threshold; with running economy considered key for overall distance running performance [3,4,5,6,7]. Additionally, monitoring the Respiratory Exchange Ratio (RER) and blood lactate values during aerobic performance testing allows one to record the interplay of the metabolic pathways that sustain performance during protracted high-intensity submaximal exercise and to provide an additional evaluation of functional fitness [8,9].

Variations in the running slope have often been employed in the continuous running literature to modify metabolic load. To increase overall exercise load, uphill running has been used as a resistance to movement method [10]. On the contrary, a downhill slope, which decreases overall metabolic load, has been explored as a means of assistive movement [11]. Downhill running allows one to place further emphasis on the eccentric phase of the muscle stretch-shortening cycle (SSC) [12] of some key muscle groups. This opens interesting possibilities; however, downhill running has not been as extensively studied as uphill running, perhaps due to concerns about possible muscle damage. It is appreciated that non-muscle-damaging exercise causes a mild, monophasic alteration in blood REDOX indicators. On the other hand, muscle–damaging exercise, especially via intense, isolated, eccentric muscle actions, causes pronounced and biphasic blood REDOX alterations [13]. These are accompanied by delayed muscle soreness and other functional or biochemical disturbances. However, despite the probability that running on sloping surfaces could differentiate REDOX responses to exercise, we found only two studies that monitored oxidative stress markers in humans; one following a post-exercise recovery timeline of up to 96 h after a submaximal downhill continuous run vs. level running [13]; and another examining, for up to 20 min, the aftermath of continuous downhill run vs. continuous uphill walking of similar energy cost [14].

Although both level and uphill HIIT improve time to exhaustion [15] and running economy [16], downhill interval running has not been systematically studied within the context of aerobic performance parameters. Moreover, to our knowledge, no study has examined exercise-induced REDOX status alterations following interval training involving either uphill or downhill slopes. We thus examined whether aerobic performance and REDOX balance may be changed when introducing a positive vs. a negative slope over an 8-week application of HIIT.

## 2. Materials and Methods

### 2.1. Participants and Study Design

Power analysis was performed using the open-source software G*Power (3.1.9.2) to calculate the minimum number of participants required to achieve a moderate effect size (η^2^ = 0.06). By defining an alpha level of 0.05, a power of 0.80, and a correlation among repeated measures of 0.5, the minimum sample size was found to be seven participants per group. Following an open call, 18 volunteers expressed interest. After applying exclusion criteria (chronic disease, including hormonal or menstrual problems, injury, medication, including use of contraception, dieting during the last 3 months) and after risks and benefits associated with participation in the study had been explained, fourteen healthy physically active adults (11M/3F) aged 31.9 ± 6.9, were selected and gave their consent to participate in the study. All participants had recreational training backgrounds (e.g., running, cycling) without formally structured training regimens and no prior involvement in HIIT. Female participants had stable cycles based on preceding calendar monitoring. The study was approved by the University of Thessaly local ethical committee (protocol number 942/10 December 2014) and was divided into three periods (Figure 1):(a)Pre-Training, when baseline data were collected on the variables of interest (including anthropometry), over a period of 8–10 days,(b)Training period, during which two training sessions per week (in total 16 sessions) were conducted over 8 weeks.(c)Post-Training, during which post-training data were collected, over a period of 8–10 days, in the same order as before, for all measurements collected during the Pre-training. For female participants, testing took place within their estimated follicular phase (between day 5 and day 11 of their cycle).

Following the Pre-Training measurements, subjects were matched for BMI and maximal oxygen uptake and were randomly divided, by a draw, into either the Uphill (UG) or Downhill group (DG). They were instructed to maintain their usual nutritional and lifestyle habits, avoid any involvement in strenuous exercise activities during the whole course of the study, and keep a diet record during a whole week, in order to replicate it during the Training Period.

During the Training Period in each of the two training sessions per week, subjects performed 10 × 30 s runs with 60 s rest in between, at 90% of MAS at either +10% (UG) or −10% (DG) grade. Those gradients and the 1:2 work-to-rest ratio were found in preliminary work by us to be well-tolerated, allowing also the completion of the uphill workout [17]. In the middle of the Training Period, a VO_2max_ test was repeated, replacing the 8th workout to re-evaluate MAS and adjust the running speed accordingly.

### 2.2. Aerobic Performance Measurements

Aerobic capacity and 90% MAS. VO_2max_ testing was conducted (at pre-, mid-, and post-training) using a progressive exercise protocol on a treadmill (Stex 8025T, Seoul, Republic of Korea) as previously described [18]. Expired gases were measured breath-by-breath with a gas analyzer (CareFusion Vmax Encore 29, San Diego, CA, USA ) and averaged every 20 s. The maximum aerobic speed (MAS) was noted at attainment of VO_2max_ [19].

Running Economy (RE). Running economy was calculated during the first four stages of a VO_2max_ test and during the first and last training sessions as oxygen consumption in relative values (VO_2_ mL·kg^−1^·min^−1^) and as oxygen consumption per unit of distance (VO_2_ mL·kg^−1^·km^−1^) [20].

Time to exhaustion at MAS (Tmax)*.* The Tmax test was conducted 48 h after the VO_2max_ test, also at level running. After a 5-min warm-up at 60% of MAS, the treadmill’s speed was quickly set to MAS, and subjects were encouraged to run until exhaustion [18].

Expired air parameters. Expired gases were also collected during the first and the last workout session, during the 10 sprint runs, and the recovery periods in between the runs, of the UG or DG training sessions, to allow for the evaluation of oxygen consumption during inclined interval exercise.

### 2.3. Blood Metabolites and REDOX Markers

Blood lactate (La). Blood lactate (La) concentration was measured in capillary blood from a fingertip with a portable analyzer (Lactate Scout, EKF Diagnostics, Cardiff, UK) at rest, after the 5th, and after the last (10th) sprint run of the first and the last workout session of the UG or DG training sessions.

Blood samples. Venous blood samples were taken at rest and within 3 min post-exercise before and after the completion of the first and the last workout of the UG or DG training sessions, using a heparinized syringe and placed into ethylene diamine tetra-acetic acid (K2EDTA)-containing tubes (Vacutainer Plus Plastic K2EDTA; Becton Dickinson, Franklin Lakes, NJ, USA). For plasma analysis, blood samples were centrifuged immediately at 1370× *g* for 10 min at 4 °C, and the supernatant was carefully collected, aliquoted in multiple Eppendorf tubes, stored at −20 °C, and thawed only once for analysis and determination of oxidative stress markers (TAC, PC, TBARS), with assays performed in triplicate.

Assays in plasma. Total antioxidant capacity (TAC) was determined spectrophotometrically, at 520 nm, according to Janaszewska and Bartosz (2002) [21], based on the scavenging of 2,2-diphenyl-1-picrylhydrazyl (DPPH) free radical. TAC values were presented as mM of DPPH reduced to 2,2-diphenyl-1-picrylhydrazine (DPPH–H).

Protein Carbonyls (PCs) were assayed spectrophotometrically, at 375 nm, as previously described by Patsoukis et al. [22], with the calculation of the Protein Carbonyl concentration being based on the molar extinction coefficient of dinitrophenylhydrazine.

Thiobarbituric-acid reactive substances (TBARS) were determined spectrophotometrically, at 530 nm, according to Buege and Aust (1978) [23]. The calculation of TBARS concentration was obtained using the molar extinction coefficient of MDA (15,600 mol/L).

### 2.4. Statistical Analyses

Data are presented as mean ± S.D. or percentage of baseline measurements. A 2 × 3 [group (uphill vs. downhill) by time (pre- vs. mid- vs. post-training)] analysis of variance (ANOVA) was used to assess VO_2max_, Respiratory Exchange Ratio (RER), and MAS.

A 2 × 3 [group (uphill vs. downhill) by time (pre- vs. mid- vs. post-training)] mixed model ANOVA was used to assess RE during the four stages of the VO_2max_ test. A 2 × 3 × 2 [group by sampling point (rest, after the 5th, and after the last repetition) by time] ANOVA was used to assess lactate concentrations during the first and last workouts. A 2 × 2 (group by time) ANOVA was used to assess Tmax. A 2 × 2 × 2 [group (uphill vs. downhill) by time (pre- vs. post-training) by sampling point (rest vs. post-exercise)] repeated measures analysis of variance (ANOVA) was used to assess oxidative stress indexes. A 2 × 2 (group by training) ANOVA was used to assess differences in percentage change from baseline values.

Moreover, for pairwise comparisons, effect size (ES) was determined by Cohen’s d (small: ≥0.2, medium: ≥0.5, and large: ≥0.80). The level of significance was set at *p* < 0.05.

## 3. Results

### 3.1. Adherence to Training

All participants (N = 14) completed the 8 weeks of training without missing any training session or any sprint within it and without reporting an adverse incident or injury. One participant in the DG missed both the post-training VO_2max_ evaluation and the Tmax test, and two participants in the UG missed the Tmax, all for personal reasons. In the related data presentation, the N value is reported accordingly.

### 3.2. Anthropometric Measurements

Body weight and BMI did not change significantly, and no differences were observed between the UG and DG either pre- or post-intervention (Table 1).

### 3.3. Aerobic Performance Measurements

VO_2max_: The two groups did not differ in terms of maximal aerobic capacity in relative or absolute values. The training intervention helped participants to maintain their VO_2max_, as there was no time effect or time by group interaction, both in absolute and relative values (Table 1).

Respiratory Exchange Ratio (RER): For RER, a significant group-by-time interaction (*p* = 0.032) was found. Post hoc tests showed that there was a significant increase in RER during the post-training VO_2max_ test only for the DG (Table 1).

MAS: For MAS, there was a significant time effect (*p* = 0.045). Although the post hoc tests did not reach conventional significance for the DG (*p* = 0.051), the effect was moderate-to-large (Cohen’s d = 0.43), suggesting a meaningful improvement (Table 1).

Running Economy: RE during submaximal level running was similar in the two groups and remained unchanged post-training; similarly, oxygen cost per unit of distance did not change (UG: 178.4 ± 27.4 vs. 180.8 ± 31.7 mL·kg^−1^·km^−1^ and DG: 212.0 ± 47.9 vs. 208.1 ± 26.9 mL·kg^−1^·km^−1^).

Oxygen cost per unit of distance, as an equivalent of the running economy, was also calculated during uphill and downhill running. This RE indicator was significantly improved after training, during the first four 30-s bouts, only in the DG (Figure 2). Thus, despite running faster after training, the downhill group had reduced oxygen cost compared to pre-training.

Time to exhaustion (Tmax): For Tmax, there was a significant group-by-time interaction (*p* = 0.021), but post hoc tests showed a non-significant tendency toward improvement in Tmax only for the Uphill group (Table 1).

### 3.4. Metabolic Load of Training

The two groups experienced significantly different metabolic load of training throughout the training intervention, as seen by the oxygen consumption and blood lactate during the first and last workout (Table 1), which was higher for the uphill group (UG).

### 3.5. Blood Metabolites and REDOX Markers

#### 3.5.1. Lactate

For lactate, there was a group-by-time-by-sample-point interaction (*p* = 0.001). Post hoc tests showed that there were significant differences between groups in post-exercise points (after the 5th and the 10th repetition), both before and after the training intervention (*p* < 0.001). Only for the uphill group, lactate concentration after the last repetition was significantly higher after the completion of the training period (Figure 3).

#### 3.5.2. TAC

For TAC, there was only a significant effect of time (*p* = 0.008). Post hoc tests revealed a significant reduction in TAC for both groups after training (*p* = 0.045, Table 2).

#### 3.5.3. PC

For PC, the analysis showed a significant sampling point effect (*p* = 0.001), a significant time by point interaction (*p* = 0.003), and a significant group effect (*p* = 0.049). Resting PC levels were similar in the two groups before training. However, resting PC levels increased after training in the Downhill group. There was an acute increase in PC levels after exercise in both groups. The acute increase in PC was lower after training for both groups, more so in the Downhill group, where there was no increase in PC after training (Table 2).

#### 3.5.4. TBARS

For TBARS, the analysis showed a significant effect of time (*p* = 0.014) and a significant point by group interaction (*p* = 0.007). Post hoc tests showed that the two groups had different resting values only after the training intervention, while post-exercise values increased for both groups after the training intervention (see Table 2).

## 4. Discussion

To our knowledge, this is the first study that compared the effects of positive vs. negative running slopes, during the implementation of interval training, on endurance performance determinants. Also, this is the first study that provides data on how slope may modulate both the acute response to HIΙΤ and the long-term adaptation to exercise-induced REDOX disturbances. Our intention was to explore how the different mechanical and metabolic loads might modulate interval running training effects. It appears that (a) peak aerobic performance was maintained for both groups despite the brevity of the intervention, (b) MAS improved in the downhill group (DG), and Tmax and lactate tolerance were improved in the uphill group (UG), and (c) the overall antioxidant capacity and post-exercise PC concentration were attenuated. Thus, it appears that changing the slope of the running surface during training can differentiate the physiological and performance outcomes of HIIT applications. The high metabolic and concentric load of the uphill training seems to promote endurance and fatigue resistance, whereas the accentuated eccentric component of the downhill training appears to promote neuromuscular adaptations that improve speed and running economy. Both approaches, however, allowed participants to maintain their peak aerobic performance.

### 4.1. Aerobic Performance and Running Economy

One key finding of this study was the maintenance of maximum oxygen consumption levels for all participants despite the brief training intervention and the low frequency of the brief training sessions. The preserved level of VO_2max_ for both the uphill and downhill groups is in accordance with results from other studies, lasting for four weeks, that found attenuation or no significant change in the VO_2max_ of runners after a level running training program with interval training that can be classified as HIIT (95–100% MAS), [24,25]. Regarding the UG, our findings also agree with studies implementing interval training with supramaximal velocities and/or extra loading (such as uphill running) [15,26,27]. Regarding the DG, our results are in line with those studies following a training program that implemented long (≤3 min) running intervals, or continuous running on negative incline (−5%, −10%) gradient for eight or five weeks [28,29], or continuous running on negative incline (−5%, −15% gradient) for four weeks [30]. Interestingly, by the end of the training intervention, the DG significantly improved MAS, without an alteration in VO_2max_ values. This observation agrees with literature showing that MAS can be improved after a level-running interval training program even in the absence of a VO_2max_ improvement [31].

It is worth noting that although both groups showed a tendency for improvement since their first VO_2max_ assessment, only the downhill group continued to improve after the mid-study VO_2max_ test. Given that the average VO_2max_ remained essentially unchanged in both groups, the continuous improvement in MAS in the downhill group suggests that neuromuscular adaptations may explain this outcome, in line with findings from accentuated eccentric load interventions [32]. Indeed, a previous study showed improvements in vertical jump and rate of force development (RFD) during a knee extension MVC following downhill interval training [33]. Secondly, the DG participants were able to reach a higher value of RER at their last VO_2max_ test, suggesting increased reliance on anaerobic metabolism, as also implied by the higher blood lactate concentration after the test, in concert with outcomes from accentuated eccentric load interventions [32].

Regarding running economy, no significant changes were observed in the two groups during level running, but economy during inclined running improved only for the DG. Results of previous studies utilizing uphill, interval training are rather controversial. One study demonstrated improved running economy during level running in well-trained runners after uphill interval training at 120% MAS [26]. Other studies demonstrated unchanged running economy at submaximal intensities (i.e., 60% and 80% of the lactate threshold) in well-trained runners after a combination of uphill interval and flat continuous training [16,28]. Notably, the aforementioned studies used a wide range of relative intensities (80–120% of MAS, 4% to 18% gradient) and different work-to-rest ratios, in participants of different fitness levels. Still, the marked improvement in inclined RE only in the DG, during the first four exercise bouts of downhill running (see Figure 2), may indicate neuromuscular adaptations involving elastic energy storage and a more effective use of the SSC in the first repetitions [7], which diminished as fatigue occurred [34]. Excessive fatigue and training overload during uphill interval training may have prevented any improvements in RE in the UG, and it remains to be clarified whether economy during level running can be improved by further manipulation of training components of uphill interval training (e.g., slope, intensity, and volume) [35].

Notably, while VO_2max_ did not change, the time to exhaustion at MAS improved by 26.4% for the uphill group. The improved time to exhaustion without concomitant VO_2max_ or running economy improvement has been previously reported following HIIT on uphill interventions [15] and was explained by increased neural input or muscle power. While this Tmax improvement for the UG was rather important and meaningful from a coaching perspective, it failed to reach statistical significance, possibly due to the small sample size and inter-individual variability. Nevertheless, in another report by us, such training modality improved fatigue tolerance, exhibited as an increased number of repetitions and total work during an isokinetic fatigue protocol [33]. Moreover, we observed an enhanced lactate tolerance, indicated by the increased post-exercise lactate concentration after training in the UG. These observations point to peripheral muscular adaptations towards more fatigue-resistant fiber properties after HIIT, as it has been previously proposed [36]. Given that MAS improved in mid-training, the use of different running speeds during post-training tests does not allow us to draw clear conclusions. Unfortunately, due to the blinding of the analyses, participants were prevented from performing an additional run-to-exhaustion test at their initial MAS and from directly comparing pre- and post-Tmax responses. Nevertheless, the considerable improvement in time to exhaustion observed in the DG is remarkable, considering the minimal time investment required for this gain, and warrants further research for future applications.

### 4.2. REDOX Responses and Adaptations to Inclined HIIT

Regarding REDOX status, the two groups showed no differences in baseline resting values in all three REDOX status indices. Concerning Total Antioxidant Capacity (TAC), the two groups were not significantly different before or after the training intervention. Although previous studies reported elevated resting TAC blood values after a short-term (three weeks) HIIT cycling training program [37], our results are in accordance with other studies that found no change or even decrements in TAC after HIIT and/ moderate intensity training programs [38,39]. Elevated TAC resting values have been related to training status in non-elite athletes and sedentary individuals [40]. On the other hand, attenuated resting TAC values have been explained on the basis of a higher pre-training baseline TAC value and fluctuations towards an optimal level as discussed in Vezzoli et al. [38]. Although the overall effect of training on TAC remains rather elusive [41,42], an inverse relationship between resting blood TAC values and training status and performance test has been documented in elite athletes [43]. We found no correlation between TAC and aerobic performance in our study. Notably, attenuated resting TAC values have been implicated as a marker of training overload [44,45] or intensified training [46]. When taken altogether, the attenuation of resting TAC values could be the outcome of either a cumulative effect of overload imposed on the training groups (eccentric component or supramaximal metabolic load), an argument discussed earlier, or of a muscle fiber shift towards fast-twitch fiber characteristics, which are more susceptible to oxidative stress [47], as also discussed earlier.

After the 8-week training intervention, resting blood Protein Carbonyl (PC) values were significantly higher only for the downhill group. However, the two groups did not differ significantly in resting PC levels, both before and after the training intervention. Our results for the UG are in accordance with previous studies, showing unaltered resting levels of PC after a period of either HIIT or a training macrocycle [37,38,40,46]. For the UG, the increased resting blood PC levels may be attributed to the preceding training session, 48 to 72 h before the last resting blood sampling took place. Aerobic exercise with an eccentric component, such as downhill running, has been shown to elevate post-exercise PC blood concentration lasting for up to 96 h after continuous downhill running [13]. Thus, it is difficult to delineate acute from chronic effects, especially given that after the last DG training session, there were no exercise-induced alterations in resting PC levels. The post-exercise PC values at the pre-training time point were significantly elevated in both groups, more so for the UG. The higher PC rise in UG was expected, since the increase in PC depends on exercise intensity [48]. After the training intervention, the exercise-induced increase in PC values was attenuated, especially so for the downhill group. This is an interesting outcome when considering again the minimum time investment of the training protocol and the minimum effort implemented by the downhill group. Overall, our data support the notion that training reduces post-exercise PC concentration [49,50] as a result of a more effective removal of oxidized proteins from circulation [38].

Thiobarbituric acid reactive substances (TBARS) resting values remained unchanged for both groups, despite a tendency for the uphill group to present elevated resting TBARS post-training. These results are in accordance with previous reports, documenting unaltered resting levels of TBARS after an eight-week period of either HIIT on a bicycle [37], or after a training macrocycle of young and adult track and field athletes [40], or after a four-week overloaded training period of triathletes [45]. Unfortunately, we were not able to find a study that used a downhill training protocol similar to ours in order to compare our results. Post-exercise TBARS values did not differ from their corresponding resting values in any of the groups. Our results are in accordance with other studies that found no change in post-exercise TBARS values after HIIT exercise on a level treadmill [51] or after a 45-min downhill treadmill run [52]. After the HIIT intervention, TBARS were elevated post-exercise vs. resting, only in the downhill group. Nevertheless, it should be noted that the DG showed lower resting values compared with the UG. Such a finding is in contrast with studies showing post-training attenuation of post-exercise TBARS levels in subjects who trained with moderate intensity, but different exercise modality (i.e., continuous training) [38]. However, we found no other research that studied the long-term effects of interval downhill running on TBARS, and hence, no direct comparisons with the literature can be performed. Nevertheless, given the fact that in a recently published study, TBARS have been shown as a reliable marker for reflecting physiological stress and recovery [53], and after considering the physiological adaptations and the resting TAC values of the downhill group as discussed earlier in the text, the cumulative effect of the eccentric component of the downhill running should be further researched.

### 4.3. Limitations

In the present study, we must acknowledge, along with some significant strengths, some unforeseen or unavoidable limitations that may hinder the generalizability of our findings. The small and unequal number of females in the two groups was a result of recruitment criteria. However, all three female participants were measured within their estimated follicular phase in order to minimize possible variations due to the menstrual cycle phase. We instructed our subjects to use diet records in order to replicate their nutritional intake before testing, but did not provide meals. Thus, subjects made their own nutritional choices, which might have influenced substrate utilization and biochemical measurements. Future work should consider a stricter diet control. The total number of participants, while being the minimal required to provide sufficient power, due to the imbalanced sex distribution and a potential learning/familiarization effect, may have affected the magnitude of the results and the overall generalizability of our findings. Still, we consider that the direction of changes measured was not affected. Future work should adopt a balanced gender distribution to allow for examining possible gender effects during inclined HIIT. Also, due to blinding procedures, our study participants did not repeat their run to exhaustion at their initial MAS; thus, our DG participants performed their post-training Tmax test at a higher speed than pre-training, clouding the interpretation of the effects of downhill interval training on time to exhaustion. This study was the first to assess acute and chronic REDOX alterations in uphill vs. downhill HIIT. However, our examination of REDOX status changes was limited to only three blood markers, which, together with limited time points of sampling, inadvertently limit the mechanistic interpretation of REDOX responses. The selected blood sampling time points (pre-exercise and 3 min after the last bout of exercise) have not permitted the appreciation of a possibly differentiated delayed modulation of the REDOX indices studied. Further work, with larger sample sizes, is needed to clarify if there is a slope dependence of the post-exercise REDOX status over 24 or 48 h.

### 4.4. Practical Implications

Based on our results, it can be substantiated that changing the slope of the running surface can differentiate the physiological and performance outcomes of HIIT applications. These findings are of practical significance for the sports scientist or trainer. Especially in the case of the working or traveling athlete, the maintenance of maximal aerobic ability with a brief HIIT intervention of running at an incline holds great usefulness. Moreover, accentuating the eccentric component of the SSC of running by downhill inclines might be used to support speed development without sacrificing aerobic ability and running economy, at least for time periods over 4 weeks but less than 8 weeks.

## 5. Conclusions

Notably, VO_2max_ was maintained after both the uphill and downhill HIIT interventions, despite the small volume and frequency of training (twice per week, with a total exercise volume of 5 min at 90% MAS for 8 weeks). From our data, it appears that slope modulation can yield physiologically meaningful results of potential interest to sports scientists and coaches: namely, uphill HIIT may serve as a training method to enhance time to exhaustion, while downhill running may improve running economy. Overall, 8 weeks of HIIT on either negative or positive slopes attenuated overall antioxidant capacity, but at the same time, reduced post-exercise PC concentration. A shift in resting TAC levels and elevated post-exercise TBARS levels warrant further investigation of the effects of HIIT on sloping surfaces on REDOX disturbance and possible links to the overloading phenomenon.

## Figures and Tables

**Figure 1 jfmk-10-00460-f001:**
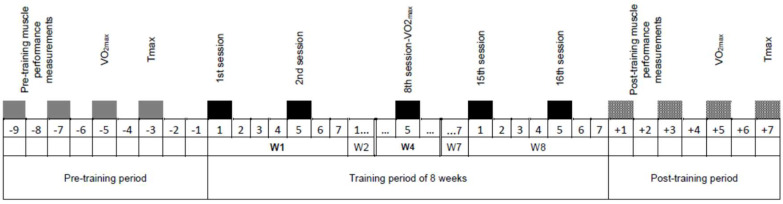
Timeline of measurements before, during, and after the 8-week intervention period. Key: Maximal aerobic capacity evaluation (VO_2max_). Test to exhaustion at MAS (Tmax), Weeks (W), Days are indicated by numbers, with the minus sign (−) indicating pre-training data collection days and the plus sign (+) indicating post-training data collection days. The double line in the W4 box indicates that the time scale is not fully depicted for all 8 weeks.

**Figure 2 jfmk-10-00460-f002:**
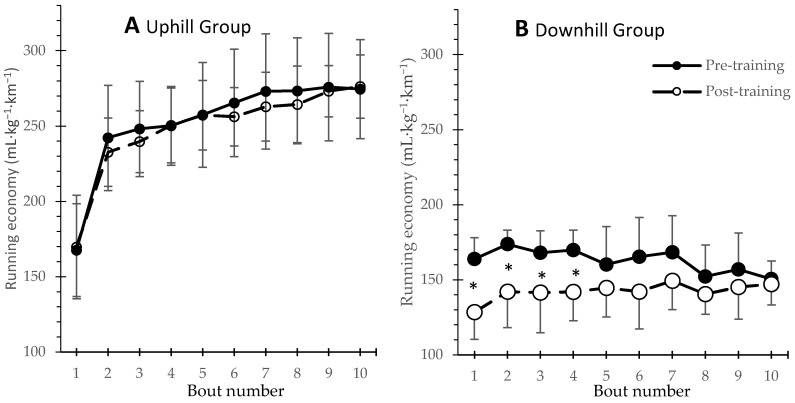
Running economy expressed as oxygen cost at each 30 s bout per unit of distance (mL·kg^−1^·km^−1^) during the first and the last training session for the Uphill (N = 7, panel **A**) and the Downhill (N = 7, panel **B**) groups (* *p* < 0.05).

**Figure 3 jfmk-10-00460-f003:**
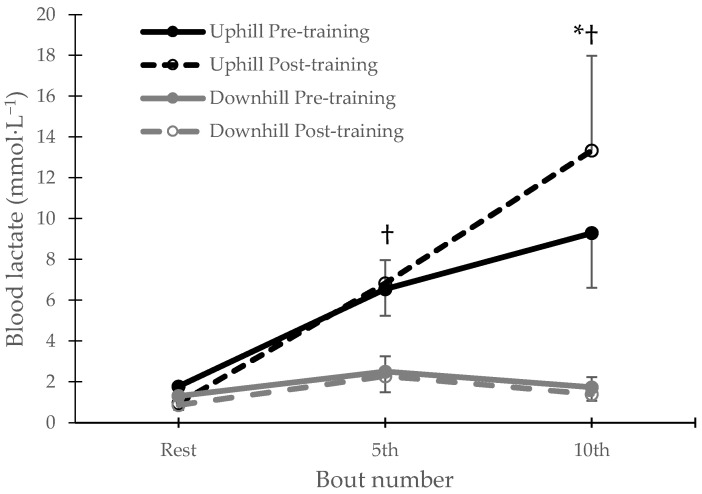
Blood lactate concentration (mmol·L^−1^) at rest, after the 5th and after the last (10th) repetition, before and after the training intervention, * denotes significantly different from the corresponding pre-training values, † denotes significantly different between groups (*p* < 0.05).

**Table 1 jfmk-10-00460-t001:** Maximal oxygen update (VO_2max_) in absolute and relative values, Respiratory Exchange ratio (RER) during the level running VO_2max_ test, Maximum Aerobic Speed (MAS), and level running Tmax before, mid, and after the training intervention for the two groups, and metabolic load of training (VO_2_) during the first and last interval training workout.

Variable	Uphill				Downhill			
	Pre-Training	Mid-Training	Post-Training	d (Pre–Post*)*	Pre-Training	Mid-Training	Post-Training	d (Pre–Post)
Weight (kg)	81.3 ± 12.7		80.5 ± 10.2	0.07	82.9 ± 23.8		80.9 ± 20.8	0.08
BMI (kg/m^2^)	26.6 ± 3.3		26.2 ± 2.4	0.12	25.1 ± 3.6		24.5 ± 2.9	0.17
VO_2max_ (L/min)	3.62 ± 0.41	3.55 ± 0.65	3.53 ± 0.66	0.16	3.55 ± 0.72	3.47 ± 0.77	3.60 ± 0.83 ^b^	0.06
VO_2max_ (mL/kg/min)	45.3 ± 5.3	43.9 ± 5.5	44.3 ± 6.5	0.16	43.6 ± 6.4	42.5 ± 5.0	44.9 ± 5.4 ^b^	0.20
RER	1.07 ± 0.06	1.11 ± 0.05	1.08 ± 0.06	0.14	1.06 ± 0.07	1.07 ± 0.05	1.12 ± 0.03 ^a^	0.91
MAS (km/h)	15.5 ± 2.3	15.8 ± 2.0	15.4 ± 2.1	0.03	13.8 ± 1.9	14.4 ± 2.3	14.5 ± 1.9	0.43
Tmax (s)	289.6 ± 89.5		354.0 ± 83.9	0.67	360.2 ± 65.7		311.0 ± 27.4	0.90
VO_2_ during training (mL/kg/min)	28.4 ± 5.4		27.9 ± 4.2	0.08	14.8 ± 1.5 ^c^		13.9 ± 1.9 ^c^	0.51

^a^ denotes significantly different from pre-training values, ^b^ denotes significantly different from mid-training values, ^c^ denotes significantly different between groups (*p* < 0.05).

**Table 2 jfmk-10-00460-t002:** Total Antioxidant Capacity (TAC), Protein Carbonyls (PCs), Thiobarbituric acid (TBARS), before and after the training intervention, and percentage change in values.

Index	Uphill (n = 7)						Downhill (n = 7)					
	Pre-Training			Post-Training			Pre-Training			Post-Training		
	Rest	Post Exercise	% Change	Rest	Post Exercise	% Change	Rest	Post Exercise	% Change	Rest	Post Exercise	% Change
TAC * (mmol DPPH·L^−1^)	0.92 ± 0.06	0.98 ± 0.08	6.10 ± 0.07	0.88 ± 0.10	0.87 ± 0.08	−1.50 ± 0.09	0.96 ± 0.10	0.92 ± 0.89	−3.83 ± 0.09	0.86 ± 0.08	0.89 ± 0.11	3.05 ± 0.09
PC (nMol·mg^−1^ protein)	0.57 ± 0.25	1.03 ± 0.17 ^b^	79.55 ± 33.06	0.68 ± 0.11	0.86 ± 0.19 ^b^	25.20 ± 75.03	0.47 ± 0.25	0.79 ± 0.72 ^ab^	69.31 ± 0.18	0.72 ± 0.18 ^c^	0.72 ± 0.19	−0.64 ± 0.18 ^ac^
TBARS * (μMol·L^−1^)	5.83 ± 1.49	5.56 ± 1.04	−4.62 ± 1.26	6.52 ± 1.16	6.23 ± 1.14 ^c^	−4.36 ± 1.15	4.97 ± 1.04	5.14 ± 1.24	3.43 ± 1.14	4.92 ± 0.84 ^a^	6.02 ± 1.26 ^b^	22.14 ± 1.05 ^ac^

* denotes main effect of training, ^a^ denotes significantly different between groups, ^b^ denotes significantly different from corresponding resting values, ^c^ denotes significantly different from corresponding pre-training values (*p* < 0.05).

## Data Availability

The original contributions presented in this study are included in the article. Further inquiries can be directed to the corresponding author.

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
