# Peer review of "Physiological, Performance, and Oxidative Stress Responses to High-Intensity Uphill and Downhill Interval Training"

_jfmk, 2025, doi:10.3390/jfmk10040460_

Round 1

Reviewer 1 Report

Comments and Suggestions for Authors

The manuscript follows a standard IMRaD format and provides a comprehensive description of methods, data, and results. The introduction successfully contextualizes the study within existing literature but tends to be overly verbose. Multiple paragraphs repeat background concepts on HIIT, REDOX physiology, and slope mechanics, diluting the focus on the specific knowledge gap being addressed. Streamlining this section—especially the redundant historical overview of HIIT—would improve readability and direct the reader more efficiently toward the study’s rationale.

The methodology is detailed and replicable, yet sometimes too narrative. Subsections (particularly in “Aerobic Performance Measurements”) include procedural redundancies and could benefit from concise phrasing. It would help to summarize repeated equipment details (e.g., gas analyzer brand, treadmill model) and integrate them in a single methodological statement rather than throughout multiple subsections. Including a brief participant flow diagram and consolidating the three “training phases” into a simpler schematic would further improve visual clarity.

The results are presented systematically and supported by quantitative data. However, the section suffers from excessive narrative explanations and redundant restatements of statistical outcomes already visible in tables and figures. For example, the description of VO₂max, MAS, and RER findings could be condensed to highlight only statistically or clinically meaningful differences. Avoid reporting every p-value inline when they are already included in tables—this contributes to clutter.

Figures and tables are appropriate, though Figure 2 lacks sufficient labeling clarity; its panels (A, B) should include descriptive subtitles (e.g., “Uphill Group” and “Downhill Group”) and more visible legends. Tables 1 and 2 contain an overwhelming amount of numerical data and abbreviations. Simplifying the layout—perhaps by removing redundant statistics such as repeated effect sizes when trivial—would increase interpretability.

Additionally, the manuscript occasionally conflates trends (“tendency for improvement”) with significant findings. The text should avoid interpretive speculation on p-values near the 0.05 threshold unless supported by a clear effect size rationale.

The discussion is thorough and demonstrates the authors’ strong command of the relevant literature. Nonetheless, it is overly long, often revisiting results already described in the results section. To enhance conciseness, paragraphs should focus on key findings: (1) maintenance of VO₂max, (2) slope-specific adaptations (improved MAS in downhill, increased Tmax and lactate tolerance in uphill), and (3) distinct REDOX responses.

Several sentences are excessively cautious or repetitive (e.g., “our findings agree with previous reports…” appears multiple times in near-identical phrasing). Reducing such repetition will significantly improve flow. The section could also benefit from a clearer mechanistic synthesis—summarizing how eccentric versus concentric load differences may explain the observed divergent metabolic and oxidative responses—before moving into literature comparisons.

A notable strength lies in the authors’ recognition of study limitations (small sample size, sex imbalance, timing of blood sampling). Still, the limitation discussion should be moved to a distinct paragraph and rephrased more assertively. Statements like “we did not get the opportunity” should be replaced with formal phrasing (e.g., “post-training testing at adjusted MAS precluded direct comparison of pre–post Tmax responses”).

The study contributes valuable evidence regarding how slope direction modulates oxidative stress and metabolic adaptation to HIIT. The findings that downhill training improves running economy and MAS while uphill training enhances fatigue tolerance are physiologically meaningful and of potential interest to sports scientists and coaches. The REDOX findings—particularly the attenuation of post-exercise PC levels despite decreased TAC—offer an intriguing biochemical dimension deserving further exploration.

However, the small sample size (n=14) and limited control for confounding variables (e.g., diet, sex distribution) should be more explicitly acknowledged as limiting the generalizability of the results. Furthermore, the mechanistic interpretation of REDOX responses would benefit from a more cautious tone, emphasizing the need for future studies with larger cohorts and extended recovery measurements.

Recommendations:

  1. Condense the introduction to two focused paragraphs that:

    • Summarize the rationale for studying slope-specific HIIT.

    • Clearly define the research hypothesis and novelty (first to assess REDOX alterations in uphill vs. downhill HIIT).

  2. Simplify and unify methodological descriptions, removing redundant procedural details and moving instrument brands to parentheses or a single sentence.

  3. Revise the results section for brevity, focusing on major statistically significant findings and eliminating repetitive explanations.

  4. Reorganize the discussion around thematic subsections (aerobic performance, REDOX adaptations, limitations, and practical implications), reducing literature repetition.

  5. Edit for language quality and consistency of terminology, correcting typographical issues, and ensuring clarity in units and abbreviations.

  6. Shorten the conclusion to a concise synthesis highlighting the distinct training adaptations and practical relevance for performance optimization.

Comments on the Quality of English Language

The English language is generally intelligible but requires editing for fluency and syntax. Common issues include:

  • Repetitive use of connectors (“while,” “however,” “moreover”) leading to overly long sentences.

  • Inconsistent tense use between past and present when describing results.

  • Overuse of abbreviations (e.g., TAC, PC, TBARS) without redefinition in late sections, which may disrupt readability for non-specialist audiences.

  • Occasional typographical errors (“HIΙΤ” with Greek letter, “mmοl/L,” and inconsistent use of commas/decimal points in numbers).

Professional English editing is strongly recommended to ensure linguistic consistency and conciseness throughout the text.

Reviewer 2 Report

Comments and Suggestions for Authors

Thank you for the opportunity to review your manuscript comparing uphill versus downhill high-intensity interval running and its effects on performance and REDOX indices. The study addresses a timely and practically relevant question, and I appreciate the careful laboratory work and clear motivation for including oxidative-stress markers alongside performance outcomes. My comments below are intended to be constructive and to help strengthen the scientific rigor, transparency, and overall clarity of your work.

Participant characteristics and sex/menstrual control: The sample (11 men / 3 women) is imbalanced. Given known sex- and cycle-phase influences on oxidative stress and substrate use, please report whether menstrual phase and/or hormonal contraceptive use were recorded/controlled and consider adding sex (and if available, cycle status) as covariates or sensitivity analyses. If not available, acknowledge as a limitation.

Expand the limitations to include: sex imbalance and absent menstrual-phase control; brief REDOX sampling window; small sample size with limited power for between-group interactions; potential learning/familiarization effects; and the mix of level- vs slope-based RE assessments.

Round 2

Reviewer 1 Report

Comments and Suggestions for Authors

Dear authors,

The draft was substantially improved. Congrats. 

These are not mandatory but could further polish the submission before final acceptance:

Statistical detail consistency: Some results still report “marginally significant” (e.g., p = 0.051) wording; replacing with “trend toward significance” or simply reporting the p-value alone would maintain full neutrality.

Figures: Ensure Figure 2 and 3 legends explicitly include units and abbreviations (e.g., ml·kg⁻¹·km⁻¹, mmol·L⁻¹).

Reference numbering: Cross-check all updated citations (since introduction trimming and reordering may have shifted numbering).

Limitations paragraph: It is strong, but one concise closing sentence linking these limitations to “future larger trials” would emphasize translational continuity.
